# Acid-base and metabolic parameters and cerebral oxygenation during the immediate transition after birth—A two-center observational study

Christian Mattersberger[1], Nariae Baik-Schneditz[1,2], Bernhard Schwaberger[1,2], Georg M. Schmölzer[3,4], Lukas Mileder[1,2], Berndt Urlesberger[1,2], Gerhard Pichler[1,2]*

1 Department of Paediatrics and Adolescent Medicine, Division of Neonatology, Medical University of Graz, Graz, Styria, Austria, 2 Research Unit for Neonatal Micro- and Macrocirculation, Department of Paediatrics and Adolescent Medicine, Medical University of Graz, Graz, Styria, Austria, 3 Centre for the Studies of Asphyxia and Resuscitation, Royal Alexandra Hospital, Edmonton, Canada, 4 Department of Paediatrics, University of Alberta, Edmonton, Canada

* gerhard.pichler@medunigraz.at

## Abstract

### Objective

The association between blood glucose level and cerebral oxygenation (cerebral regional oxygen saturation [$crSO_2$] and cerebral fractional tissue oxygen extraction [FTOE]) in neonates has already been described. Aim of the present study was to investigate if acid-base and other metabolic parameters have an impact on cerebral oxygenation immediately after birth in preterm and term neonates.

### Study design

Post-hoc analyses of secondary outcome parameters of two prospective observational studies were performed. Preterm and term neonates born by caesarean section were included, in whom i) cerebral near-infrared spectroscopy (NIRS) measurements were performed during the first 15 minutes after birth and ii) a capillary blood gas analysis was performed between 10 and 20 minutes after birth. Vital signs were routinely monitored with pulse oximetry (arterial oxygen saturation [$SpO_2$] and heart rate [HR]). Correlation analyses were performed to investigate potential associations between acid-base and metabolic parameters (lactate [LAC], pH-value [pH], base-excess [BE] and bicarbonate [$HCO_3$]) from capillary blood and NIRS-derived $crSO_2$ and FTOE at 15 minutes after birth.

### Results

One-hundred-fifty-seven neonates, 42 preterm neonates (median gestational age [IQR] 34.0 weeks [3.3], median birth weight 1845g [592]) and 115 term neonates (median gestational age [IQR] 38.9 weeks [1.0], median birth weight 3230g [570]) were included in the study. Median $crSO_2$ [IQR] values at 15 minutes after birth were 82% [16] in preterm neonates and 83% [12] in term neonates. Median FTOE [IQR] values at 15 minutes after birth were 0.13 [0.15] in preterm neonates and 0.14 [0.14] in term neonates. In preterm neonates,

**Data Availability Statement:** All relevant data are within the paper and its Supplemental Information files.

**Funding:** GP is a recipient of the Culture Department of the City of Graz, Austria. GMS is a recipient of the Heart and Stroke Foundation/ University of Alberta Professorship of Neonatal Resuscitation, a National New Investigator of the Heart and Stroke Foundation Canada and an Alberta New Investigator of the Heart and Stroke Foundation Alberta. This research has been facilitated by the Women and Children's Health Research Institute through the generous support of the Stollery Children's Hospital Foundation.

**Competing interests:** The authors have declared that no competing interests exist.

**Abbreviations:** BE, base excess; $CO_2$, carbon dioxide; $crSO_2$, cerebral regional oxygen saturation; FTOE, fractional tissue oxygen extraction; $HCO_3$, bicarbonate; HR, heart rate; LAC, lactate; NIRS, near-infrared spectroscopy; pH, pH value; $SpO_2$, arterial oxygen saturation.

higher LAC and lower pH and BE were associated with lower $crSO_2$ and higher FTOE. In term neonates, higher $HCO_3$ was associated with higher FTOE.

## Conclusion

There were significant associations between several acid-base and metabolic parameters and cerebral oxygenation in preterm neonates, while in term neonates only $HCO_3$ correlated positively with FTOE.

## Introduction

The transition from intra- to extra-uterine life of a neonate is characterized by independence from placental oxygen and nutrient supply. Disturbances of this transition immediately after birth may lead to impaired oxygen and nutrient supply and, thus, to possible irreversible impairments. Due to its high vulnerability, the neonatal brain should be one of the target organs of clinical observation after birth. The (autoregulation) mechanism to maintain cerebral oxygen and nutrients supply, especially during and after perinatal transition, is still not completely understood [1–3]. Hyper- and hypoxia during this period can possibly lead to cerebral damages like intraventricular haemorrhage, periventricular leukomalacia or retinopathy of prematurity [4–6]. The currently recommended monitoring for the assessment of neonates after birth includes peripheral arterial oxygen saturation ($SpO_2$) and heart rate (HR), measured with pulse oximetry and/or electrocardiogram [7–9]. Unfortunately, the routinely used methods do not assess cerebral oxygen delivery and oxygen consumption and, therefore, neglect potentially crucial information regarding the cerebral oxygenation [10]. Near-infrared spectroscopy (NIRS) is a non-invasive, real-time method which enables the monitoring of cerebral tissue oxygenation and hemodynamics. Reference ranges for neonatal cerebral oxygenation (cerebral regional tissue oxygen saturation [$crSO_2$] and cerebral fractional tissue oxygen extraction [FTOE]) immediately after birth have already been established for different NIRS devices [11–13]. One two-center prospective observational study demonstrated an association between lower $crSO_2$ and intraventricular haemorrhage in preterm neonates during the immediate transition [10]. Further, intervention guidelines based on cerebral oxygenation monitoring measured with NIRS aiming at reducing the time of hypoxia during after birth have already been developed [14, 15].

Cerebral oxygenation is influenced by oxygen delivery, based on vascular resistance and cardiac output, and by oxygen consumption. Both, oxygen delivery and oxygen consumption may be influenced by the acid-base status and the metabolism. A recent study has already demonstrated a negative association between blood glucose and cerebral oxygenation in preterm and term neonates immediately after birth [16]. However, to date no data are available regarding possible associations between acid-base and metabolic parameters and cerebral oxygenation ($crSO_2$, FTOE) in neonates during immediate transition after birth.

Acid-base and metabolic parameters reflect sufficient oxygenation and the counter regulation mechanism in hypoxic conditions of the organism. During immediate neonatal transition after birth, acid-base and metabolic parameters, measured out of umbilical cord blood, have already been used as outcome predictors and as indicators for further interventions in neonates [17, 18]. A high lactate (LAC) level is an unspecific chemical marker e.g. for neonatal hypoxia and may be associated with an increased risk for adverse neurological outcome [19, 20]. Furthermore, deviations from normal acid-base levels (pH-value [pH], base-excess [BE], and bicarbonate [$HCO_3$]) during immediate postnatal transition are further predictors for

poor neurological outcome in preterm and term neonates [20–22]. However, there are currently no data available about the influence of changes in the acid-base status and the metabolism and the effect on $crSO_2$ and FTOE in neonates during immediate transition after birth. Therefore, the aim of the present study was to identify additional associations between acid-base and metabolic parameters and their relation to cerebral oxygenation in preterm and term neonates. We hypothesized that higher LAC and lower pH, BE and HCO3 suggesting impaired metabolism is associated with lower crSO2 and higher FTOE values in preterm and term neonates 15 minutes after birth.

## Material and methods

This study analysed secondary outcome parameters of two prospective observational studies, one carried out at the Royal Alexandra Hospital, Edmonton, Canada, and one at the Division of Neonatology, Medical University of Graz, Austria. The studies were conducted between February 2014 and February 2015 in Edmonton and from October 2015 to September 2018 in Graz [23].

### Inclusion and exclusion criteria

Preterm and term neonates with the decision to conduct full life support and who were born by caesarean section were included. Neonates with congenital malformations (e.g. congenital diaphragmatic hernia) were excluded. Institutional ethical approvals (Health Research Ethics Board, University of Alberta, Canada: Pro00032233; Ethical committee, Medical University of Graz, Austria: 27–465 ex 14/15) were obtained. In Graz written parental consent was obtained before birth, while in Edmonton deferred consent was used with written parental consent after delivery.

### Study procedure

The neonates were brought to the resuscitation desk and were placed under an overhead heater in a supine position. Preterm neonates <29 week of gestation were covered in a plastic wrap according the neonatal resuscitation guidelines. Resuscitation was performed by dedicated resuscitation teams (neonatologist/experienced resident and nurse), which were not involved in the study, according the neonatal resuscitation guidelines [24, 25].

Respiratory support was provided using continuous positive airway pressure or positive pressure ventilation using a T-piece device (Neopuff Infant Resuscitator, Fisher & Paykel Healthcare, Auckland, New Zealand). The level of oxygen was titrated according the neonatal rescuscitation guidelines. For $SpO_2$ and HR monitoring, a pulse oximetry probe (IntelliVue MP30 Monitor, Philips, Amsterdam, The Netherlands) was placed around the right wrist/hand. For measuring $crSO_2$ and FTOE, an INVOS 5100 monitor (Covidien, Minnesota, USA) with a neonatal sensor was applied on the left fronto-parietal head in each neonate and fixed with a cohesive conforming bandage. Cerebral oxygenation was monitored during the first 15 minutes after birth.

The multi-channel system alpha-trace digital MM (BEST Medical Systems, Vienna, Austria) was used to store all variables for subsequent analyses. Values of the non-invasive monitoring were stored every second. The 15th minute median values of each neonate were used for analyses to be closest to the mean blood sample times in the two groups. As a quality criterion, $crSO_2$ and $SpO_2$ values were eliminated if $crSO_2$ was higher than $SpO_2$ [26]. FTOE was calculated by the following formula: $([SpO_2-crSO_2]/SpO_2)$ [27].

Acid-base and metabolic parameters (LAC, pH, BE, and $HCO_3$), measured from capillary blood samples in the period between 10–20 minutes after birth using a blood gas analyser

(ABL 800 Flex, Fa. Drott, Wiener Neustadt, Austria), were used for analyses. The capillary blood samples were taken according to the discretion of the attending neonatologist.

## Statistical analysis

Data are presented as median with interquartile range (IQR).

Demographic data and measured parameters ($SpO_2$, HR, $crSO_2$, FTOE, LAC, pH, BE, $HCO_3$) of preterm and term neonates were compared. Categorical demographic variables were compared with the Chi-square test or Fisher's exact test. Continuous variables were compared using Student's t-test or Mann-Whitney-U test, as appropriate. The associations between $crSO_2$/FTOE and acid-base and metabolic parameters were analyzed using Spearman's rank correlation coefficient or Pearson's correlation, as appropriate. The correlation analyses were considered in an explorative sense; therefore, no multiple testing corrections were performed. A p-value $<0.05$ was considered statistically significant. The statistical analyses were performed using IBM SPSS Statistics 26.0.0 (IBM Corporation, Armonk, NY, USA).

## Results

Out of 500 eligible neonates, in whom NIRS measurements were performed, we included 42 preterm and 115 term neonates. Most neonates were excluded because of the lack of a blood sample or $crSO_2$ data at the 15th minute after birth (Fig 1). One hundred forty-two mothers of the included neonates received regional anesthesia, nine mothers received general anesthesia, and in six mothers anesthesia was not documented for the caesarean section. Indications for a caesarean section were previous caesarean sections (n = 38), intrauterine growth restriction (n = 10), multiple birth (n = 22), breech presentation (n = 10), suspected fetal and/or maternal infection (n = 1), transverse presentation (n = 1), placenta praevia (n = 3), pre-eclampsia and eclampsia (n = 2), premature rupture of the membrane and premature labor (n = 8), others (n = 19) and not documented (n = 43).

Demographic data of the included neonates are presented in Table 1. Besides gestational age and birth weight, there were significant differences in Apgar scores, $SpO_2$, and rectally measured central temperature between preterm and term neonates, with lower values in preterm neonates. Blood samples were taken at a median (IQR) postnatal age of 17 minutes (3 minutes) in preterm and of 16 minutes (2 minutes) in term neonates (p = .051).

During the study period, 12 preterm (28.6%) and nine term (7.8%) neonates received supplemental oxygen (p = < .001). 19 preterm (45.2%) and 19 term (7.8%) neonates received non-invasive respiratory support (p = < .001). Four preterm (9.5%) neonates, but no term neonate, were intubated and mechanically ventilated (p = .019) (Table 1).

### Cerebral tissue oxygenation

There was no significant difference in $crSO_2$ and FTOE between preterm and term neonates (Table 1). Six (14.3%) preterm neonates had $crSO_2$ values <10th centile and nine (21.4%) >90th centile at 15 minutes after birth according to published centiles by Pichler et al. [12] Four (3.5%) term neonates had $crSO_2$ values <10th centile and 24 (20.9%) >90th centile. In five (11.9%) preterm neonates FTOE values were <10th centile and in three (7.1%) >90th centile at 15 minutes after birth. Twenty-nine (25.2%) term neonates had FTOE values <10th centile and three (2.6%) >90th centile.

### Acid-base and metabolic parameters

There were a significant differences in pH and BE between preterm and term neonates, with lower values in preterm neonates.

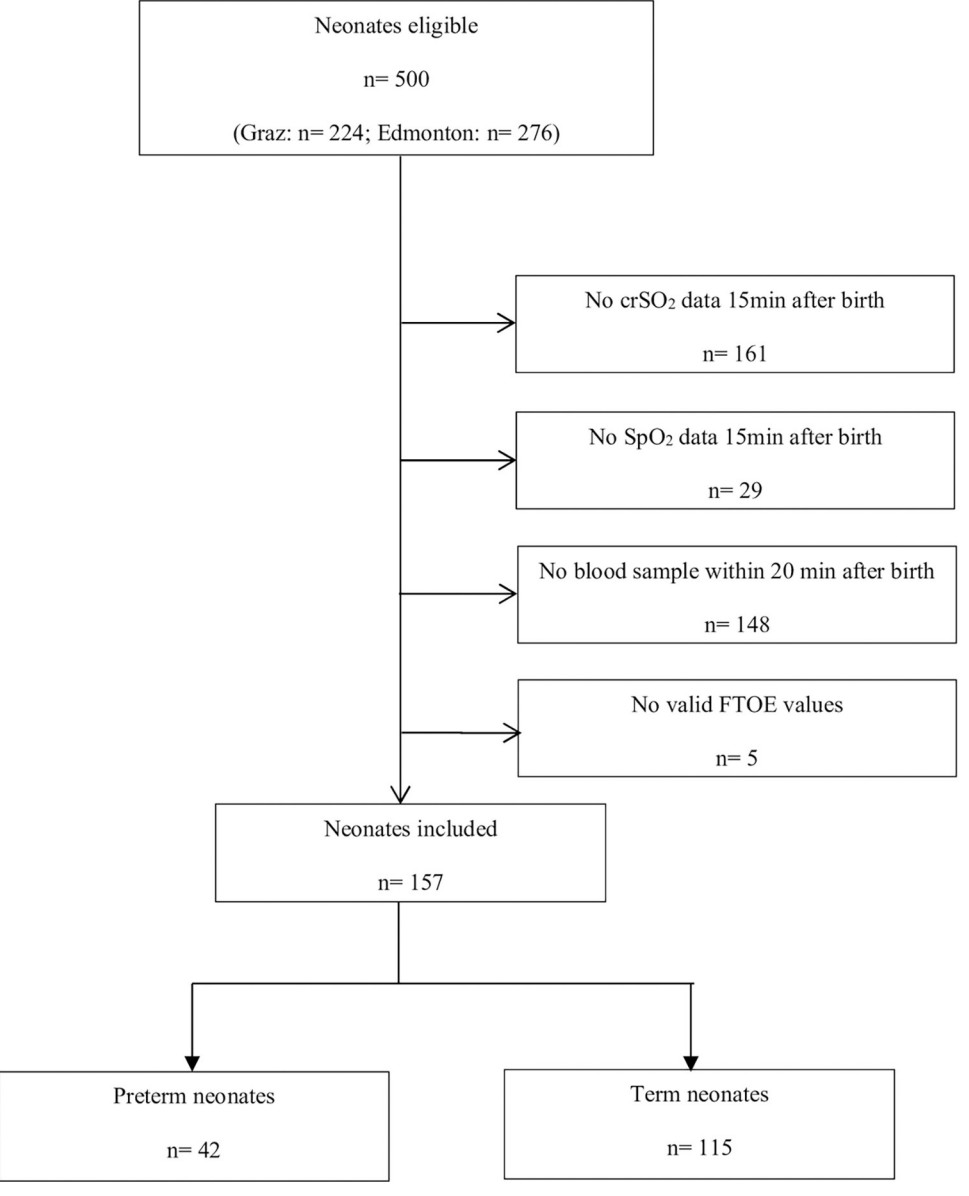

**Fig 1. Patient flow diagram.** [SpO$_2$ = arterial oxygen saturation, crSO$_2$ = cerebral regional oxygen saturation; FTOE = fractional tissue oxygen extraction].

One (0.8%) term neonate had LAC <2.5th centile and 10 (8.5%) >97.5th centile after birth according to the centiles published by Cousineau et al. [28]. Seventy-four (63.2%) term neonates had pH values <2.5th centile and no neonate had pH values >97.5th centile at 15 minutes after birth [28].

## Correlation analyses of acid-base and metabolic parameters and cerebral tissue oxygenation

Table 2 shows the correlation analyses of cerebral oxygenation and acid-base and metabolic parameters. Scatter plots of acid-base and metabolic parameters and cerebral oxygenation for preterm and term neonates are provided in Fig 2.

**Table 1. Demographic and clinical data of the study population.** Mean value and interquartile range (IQR) at the 15th minute after birth of preterm and term neonates with group comparison.

| | | Preterm n = 42 | IQR | Term n = 115 | IQR | Group comparison p-value |
|---|---|---|---|---|---|---|
| *Demographics* | Gestational age (**weeks**) | 34.0 | 3.3 | 38.9 | 1 | < .001 |
| | Birth weight (**g**) | 1845.0 | 592 | 3230.0 | 570 | < .001 |
| | Apgar **1 min** | 8 | 1 | 9 | 0 | < .001 |
| | Apgar **5 min** | 9 | 2 | 10 | 1 | < .001 |
| | Apgar **10 min** | 10 | 1 | 10 | 0 | < .001 |
| *Monitoring* | Mean arterial blood pressure (**mmHg**) * | 43 | 12 | 45 | 11 | .065 |
| | Rectal temperature (**C°**) * | 36.8 | 0.5 | 37 | 0.3 | .002 |
| | Arterial oxygen saturation (**%**) * | 95 | 7 | 96 | 4 | .004 |
| | Heart rate (**bpm**) * | 155 | 23 | 152 | 21 | .548 |
| | Cerebral regional oxygen saturation (**%**) * | 82 | 16 | 83 | 12 | .144 |
| | Fractional tissue oxygen extraction * | 0.13 | 0.15 | 0.14 | 0.14 | .333 |
| *Ventilation* | Supplemental oxygen, **n (%)** | 12 | (29) | 9 | (8) | < .001 |
| | Non-invasive respiratory support, **n (%)** | 19 | (45) | 19 | (17) | < .001 |
| | Endotracheal intubation, **n (%)** | 4 | (10) | 0 | (0) | .019 |
| *Acid-base metabolism* | Lactate (**mg/dL**) | 2.8 | 1.1 | 2.7 | 1.2 | .346 |
| | pH | 7.267 | 0.095 | 7.293 | 0.061 | < .001 |
| | Base Excess (**mmol/L**) | - 2.3 | 2.6 | - 0.9 | 2.8 | .001 |
| | Bicarbonate (**mmol/L**) | 21.0 | 2.6 | 21.6 | 1.9 | .126 |
| | Time of blood sample, postnatal (**min**) | 17 | 3 | 16 | 2 | .051 |

* = 15 minutes postnatal.

In preterm neonates, correlation analyses between crSO$_2$ and LAC showed a significantly negative correlation, while FTOE and LAC were positively correlated. In preterm neonates, correlation analyses between crSO$_2$ and pH and BE showed a significant positive correlation and FTOE was negatively correlated with pH and BE (Table 2).

In term neonates, there were no significant correlations between crSO$_2$ and LAC, pH, BE, HCO$_3$, but a significantly positive correlation between FTOE and HCO$_3$ (Table 2).

## Discussion

To our knowledge, the present study is the first to demonstrate several significant correlations between acid-base and metabolic parameters and cerebral oxygenation especially in preterm, but also in term neonates during the immediate transition after birth.

**Table 2. Spearman/Pearson correlation analyses of acid-base and metabolic parameters and cerebral oxygenation in preterm and term neonates 15 minutes after birth.**

| | | *Preterm (n = 42)* | | | | *Term (n = 115)* | | | |
|---|---|---|---|---|---|---|---|---|---|
| | | crSO2 (%) | p-Value | FTOE | p-Value | crSO2 (%) | p-Value | FTOE | p-Value |
| *Lactate (**mg/dL**)* | | $\varrho$ = -.388 | .015 | $\varrho$ = .379 | .019 | $\varrho$ = -.128 | .185 | $\varrho$ = .118 | .244 |
| *pH* | | $\varrho$ = .585 | < .001 | $\varrho$ = -.489 | .001 | $\varrho$ = -.129 | .172 | $\varrho$ = .163 | .083 |
| *Base Excess (**mmol/L**)* | | $\varrho$ = .355 | .021 | $\varrho$ = -.314 | .045 | $\varrho$ = -.144 | .128 | $\varrho$ = .179 | .058 |
| *Bicarbonate (**mmol/L**)* | | $\varrho$ = .285 | .079 | $\varrho$ = -.206 | .215 | $\varrho$ = -.173 | .069 | $\varrho$ = .216 | .023 |

$\varrho$ = Person/Spearman correlation coefficient.

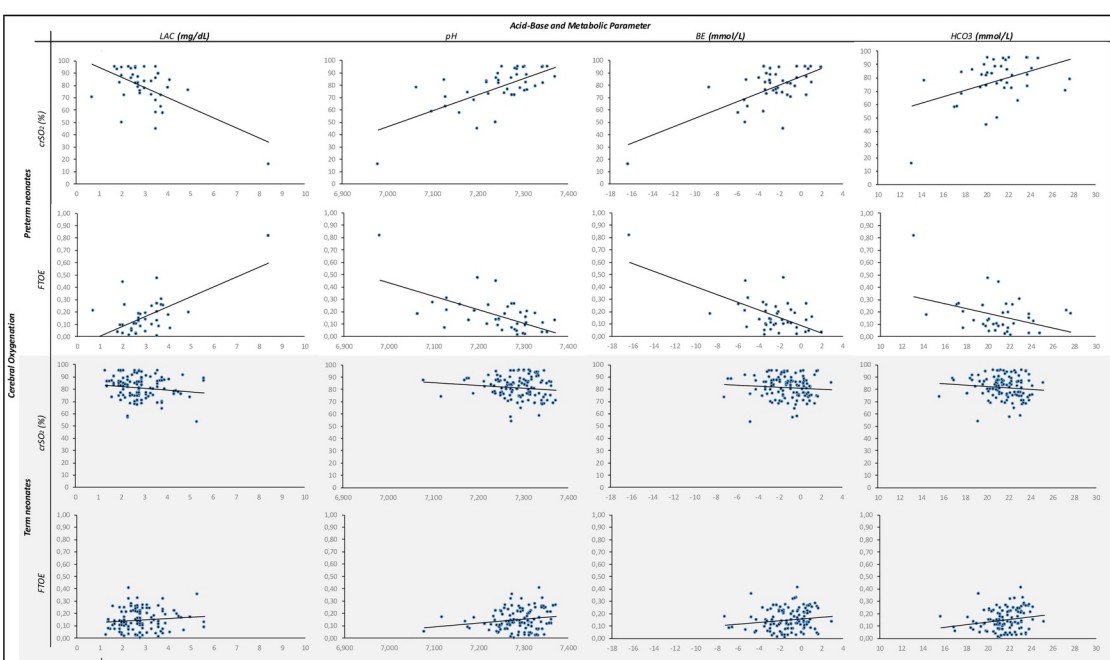

**Fig 2. Scatter plots of acid-base and metabolic parameters and cerebral oxygenation (cerebral regional oxygen saturation [crSO₂] and fractional-tissue oxygen extraction [FTOE]) in preterm and term neonates.**

Previous investigations on correlations between cerebral oxygenation and acid-base and metabolic parameters during the neonatal period were controversial. Outside the transitional period after birth, some studies demonstrated correlations between cerebral tissue oxygenation and acid-base and metabolic parameters [29–34] whereas others found none at all [35–41] in neonates. There are several possible reasons for these different findings, among them the gestational age, the postnatal age at the time of measurement, the numbers of neonates included and, therefore, the statistical power, and the timing of measurements of cerebral oxygenation and acid-base and metabolic parameters.

## Lactate and cerebral oxygenation

LAC is a metabolite within the anaerobic energy maintenance system and is increased during inadequate cellular oxygenation [42]. Our study demonstrated a negative correlation between crSO₂ and LAC and a positive correlation between FTOE and LAC in preterm, but not in term neonates during the immediate transition after birth. Similar results have been published for extremely preterm neonates during the first days after birth [33]. In contrast, other studies of critically-ill neonates and infants who had been free of cerebral disease or trauma for up to one year [35], of paediatric patients during heart surgery [31], and of neonates with prenatally diagnosed congenital heart disease during the first 72 hours after birth [39] did not demonstrate any correlations.

A possible explanation of the observed negative correlation between crSO₂ and LAC is that increase in LAC leads to a pulmonary vasoconstriction and a reduction in cardiac stroke volume as a result of reduced cardiac contractility [43] Therefore, increase in LAC levels might lead to impaired cardiac output and thus to a decrease of oxygen delivery to the brain, followed by a reduced crSO₂. Furthermore, animal studies demonstrated that with increase in LAC the pH level decreases with potential effects on the neonatal heart [43, 44]. This effect of LAC on

the neonatal heart seems to concern only preterm neonates. A recent study did not demonstrate this effect of LAC on the cardiac output in term born neonates during the first 6 hours after birth [45].

## pH and cerebral oxygenation

Both, respiratory as well as metabolic acidosis may result from inadequate oxygen delivery and impaired gas exchange. The pH level of umbilical artery blood is the most sensitive indicator of fetal hypoxia during delivery and is used for evaluating postnatal adaptation [46]. We found a positive correlation between $crSO_2$ and pH measured from a capillary blood sample 15 minutes after birth and a negative correlation between FTOE and pH in preterm, but not in term neonates during the immediate transition after birth. There was only a trend towards a positive correlation between FTOE and pH measured from a capillary blood sample in term neonates.

Similar results of a positive correlation between pH and $crSO_2$ have been published for neonates shortly before birth [29] and in neonates and infants undergoing thoracoscopic repair of congenital diaphragmatic hernia and esophageal atresia [30]. In contrast, a negative correlation between $crSO_2$ and pH was demonstrated in paediatric patients undergoing surgical heart procedures [31] and in term-born infants suffering from hypertrophic pyloric stenosis during correction of metabolic alkalosis [32]. Against them, some studies demonstrated the absence of correlations between $crSO_2$ or FTOE and pH in i) critically-ill neonates and infants who have been free of cerebral disease and trauma for up to one year [35], ii) in very preterm neonates during the first six hours of life [36], iii) in neonates and children during cardiopulmonary bypass surgery [37], iv) in extremely preterm neonates with very low birth weight during sodium $HCO_3$ infusion for correction of metabolic acidosis during the first postnatal week [38], v) in term born infants with prenatally diagnosed congenital heart disease during the first 72 hours after birth [39], vi) in clinically stable preterm infants in the intensive care unit [40] and vii) in term neonates with persisting pulmonary hypertension [41].

A possible explanation of the observed positive correlation between $crSO_2$ and pH in preterm but not in term neonates is that acidosis leads to a reduced contractility of cardiomyocytes, and to a reduced responsiveness on catecholamines [47]. Both mechanism may subsequently result in a reduced cardiac output, and thus, may lead to a decrease in oxygen delivery to the brain with reduced $crSO_2$. Though, in contrast to these laboratory findings, the myocardial contractility of hemodynamically stable preterm neonates during the transitional period remains relatively unaffected during acidosis, even at pH values below 7.00 [48]. There seems to be no relationship between pH and cardiac output in preterm neonates during the first three days after birth [48].

In addition, depending on the tissue, severe acidosis or alkalosis leads to a vasodilatation or vasoconstriction. On the one side, acidosis leads to an increased pulmonary vascular resistance and raise in pulmonary artery pressure. On the other side, acidosis leads to a redistribution of blood from peripheral veins into the lungs with consequential increase in preload and following the frank starling mechanism an increase in cardiac output [49]. In preterm neonates after the transitional period defined as days 4–14 after birth, acidosis leads to a decrease in systemic vascular resistance with subsequent increase in left ventricular output. In preterm neonates during the transitional period defined as days 1–3, there seems to be no relationship between the pH and the systemic vascular resistance. Following this, the systemic vascular response to acidosis undergoes a postnatal maturational process in preterm neonates during the first two postnatal weeks [48].

Beside the vasodilative effect of acidosis on the systemic vascular resistance, acidosis leads to vasodilation of cerebral vessels resulting in an increase in cerebral blood flow and

oxygenation [31]. Therefore one would expect a negative correlation of pH and $crSO_2$. However, we observed a positive correlation of pH and $crSO_2$. An immature compensation mechanism with inadequate vasodilation due to acidosis in preterm neonates might explain our finding.

## Base excess and cerebral oxygenation

The BE quantifies the magnitude of the metabolic acidosis and represents a risk factor for central neurologic injury and is a prognostic marker for short- and long-term outcome in asphyxiated neonates [50, 51]. In our study, BE was positively correlated with $crSO_2$ and negatively with FTOE in preterm neonates during the immediate transition after birth. Again, there was only a trend towards a positive correlation between BE and FTOE in term neonates.

Studies demonstrated a negative correlation between the cerebral oxygenation and BE i) in fetuses shortly before birth [29], ii) in term-born infants suffering from hypertrophic pyloric stenosis [32] and iii) in extremely preterm neonates receiving sodium $HCO_3$ during the first 24 hours after birth [34]. In contrast, other studies found no correlation between BE and $crSO_2$ or FTOE i) in critically-ill neonates and infants up to one year [35], ii) in extremely preterm neonates with very low birth weight during sodium $HCO_3$ therapy due to metabolic acidosis during the first week of life [38] and iii) in clinically stable preterm infants in the neonatal intensive care unit [40].

A possible explanations of our observed positive correlation between BE and $crSO_2$ in preterm neonates are the also observed correlations of $crSO_2$ with other acid-base parameters since BE is calculated out of acid-base parameters [52]. However, BE is an indicator of shock and efficacy of resuscitation and reflective of volume deficit [53]. It can be assumed that decreased BE demonstrates a centralisation with its resulting hemodynamic problems for the oxygen delivery that is in accordance with the present findings in preterm neonates.

## Bicarbonate and cerebral oxygenation

Buffers are important for proton elimination and, particularly, $HCO_3$ is the most important buffer system and contributes approximately 35% to this effect [54]. Our study showed a trend towards a positive correlation between $crSO_2$ and $HCO_3$ in preterm and term neonates. Further, FTOE correlated positively with $HCO_3$ and $crSO_2$ showed a trend towards a negative correlation to $HCO_3$ in term neonates during the first 15 minutes after birth.

Similar results have been published in extremely preterm neonates during sodium $HCO_3$ administration during the first 24 postnatal hours [34]. In contrast, another study reported a negative correlation between cerebral oxygenation and $HCO_3$ in term-born infants with hypertrophic pyloric stenosis suffering from metabolic alkalosis that is also in accordance with the present study [32]. The absence of a correlation between $crSO_2$ and $HCO_3$ has been described in extremely preterm neonates with very low birth weight and metabolic acidosis during the first week after birth [38] and in clinically stable preterm infants receiving treatment in the neonatal intensive care unit [40].

The present observations are in accordance with the findings in pH and $crSO_2$ and FTOE in term and preterm neonates, since pH and $HCO_3$ are strongly linked together.

Based on the observed multiple significant correlations between acid-base status and cerebral oxygenation especially in preterm neonates in comparison to term neonates, we hypothesize that the capacity of the cerebral autoregulation mechanism to maintain constant cerebral oxygen supply is less dependent on acid-base status with increasing gestational age. Given this assumption, an increase in gestational age obviously leads to an increased ability to

compensate an impaired cerebral oxygenation in neonates during the immediate neonatal transition after birth.

One of the strengths of this investigation is that it was a two center study with a rather high sample size especially regarding term neonates. Limitations include the smaller sample size of preterm neonates. Nonetheless, we still found significant associations in preterm neonates. A further limitation is the difference in timing between the NIRS measurements and the taking of blood samples. Nevertheless, it can be assumed that the components of the acid-base metabolism did not change significantly between NIRS measurements until minute 15 after birth and blood sample taking. Another limitation of our study are possible interactions of the investigated acid-base and metabolic parameters, which may have potentially confounded the observed effects on cerebral oxygenation. Finally, blood gas analysis use the Van Slyke equation to calculate various acid-base variables by measuring the pH, hemoglobin content and partial pressure of $CO_2$ [55]. This could be a further confounder for the observed correlation.

## Conclusion

Fifteen minutes after birth, acid-base and metabolic parameters are significantly associated with $crSO_2$ and FTOE in preterm neonates. An increase of LAC, and a decrease of pH and BE, all three are associated with a decrease in $crSO_2$ and an increase in FTOE in preterm neonates only. In term neonates there is a significant positive correlation between $HCO_3$ and FTOE. Still, the causal relationship between acid-base and metabolic parameters and cerebral oxygenation during the immediate postnatal transition, and thus potential influences on short- and long-term neonatal outcomes, have to be evaluated.

## Supporting information

**S1 Data.**
(XLSX)

## Author Contributions

**Conceptualization:** Christian Mattersberger.

**Data curation:** Christian Mattersberger, Nariae Baik-Schneditz, Bernhard Schwaberger, Georg M. Schmölzer, Lukas Mileder, Berndt Urlesberger, Gerhard Pichler.

**Formal analysis:** Christian Mattersberger, Gerhard Pichler.

**Funding acquisition:** Georg M. Schmölzer, Gerhard Pichler.

**Investigation:** Christian Mattersberger, Nariae Baik-Schneditz, Bernhard Schwaberger, Georg M. Schmölzer, Lukas Mileder, Berndt Urlesberger, Gerhard Pichler.

**Methodology:** Christian Mattersberger, Berndt Urlesberger, Gerhard Pichler.

**Project administration:** Christian Mattersberger, Georg M. Schmölzer, Berndt Urlesberger, Gerhard Pichler.

**Resources:** Georg M. Schmölzer, Berndt Urlesberger, Gerhard Pichler.

**Software:** Christian Mattersberger.

**Supervision:** Georg M. Schmölzer, Gerhard Pichler.

**Validation:** Georg M. Schmölzer, Berndt Urlesberger, Gerhard Pichler.

**Visualization:** Christian Mattersberger, Gerhard Pichler.

**Writing – original draft:** Christian Mattersberger, Gerhard Pichler.

**Writing – review & editing:** Christian Mattersberger, Nariae Baik-Schneditz, Bernhard Schwaberger, Georg M. Schmölzer, Lukas Mileder, Berndt Urlesberger, Gerhard Pichler.

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
