## [Decision Letter · Decision Letter 0]

25 Apr 2022

PONE-D-22-00810Acid-base and metabolic parameters and cerebral oxygenation during the immediate transition after birth - An two-center observational studyPLOS ONE

Dear Dr. Pichler,

Thank you for submitting your manuscript to PLOS ONE. After careful consideration, we feel that it has merit but does not fully meet PLOS ONE’s publication criteria as it currently stands. Therefore, we invite you to submit a revised version of the manuscript that addresses the points raised during the review process.

Please submit your revised manuscript by Jun 05 2022 11:59PM. If you will need more time than this to complete your revisions, please reply to this message or contact the journal office at plosone@plos.org. Please include the following items when submitting your revised manuscript:A rebuttal letter that responds to each point raised by the academic editor and reviewer(s). You should upload this letter as a separate file labeled 'Response to Reviewers'.A marked-up copy of your manuscript that highlights changes made to the original version. You should upload this as a separate file labeled 'Revised Manuscript with Track Changes'.An unmarked version of your revised paper without tracked changes. You should upload this as a separate file labeled 'Manuscript'.

We look forward to receiving your revised manuscript.

Kind regards,

Barbara Wilson Engelhardt, MD

Academic Editor

PLOS ONE

Journal Requirements:

GP is a recipient of the the Culture Department of the City of Graz, Austria. 

GMS is a recipient of the Heart and Stroke Foundation/University of Alberta Professorship of Neonatal Resuscitation, a National New Investigator of the Heart and Stroke Foundation Canada and an Alberta New Investigator of the Heart and Stroke Foundation Alberta. This research has been facilitated by the Women and Children’s Health Research Institute through the generous support of the Stollery Children's Hospital Foundation.

Reviewers' comments:

Reviewer's Responses to Questions

**Comments to the Author**

1. Is the manuscript technically sound, and do the data support the conclusions?

Reviewer #1: Partly

Reviewer #2: Partly

2. Has the statistical analysis been performed appropriately and rigorously? 

Reviewer #1: Yes

Reviewer #2: Yes

3. Have the authors made all data underlying the findings in their manuscript fully available?

Reviewer #1: Yes

Reviewer #2: Yes

4. Is the manuscript presented in an intelligible fashion and written in standard English?

Reviewer #1: Yes

Reviewer #2: Yes

5. Review Comments to the Author

Reviewer #1: It seems to be a new area to explore impact of acid-base and other metabolic parameters on cerebral

oxygenation in preterm and term neonates immediately after birth. Hence I have some observation.

Method:

Required sample size better to mention.

Result:

Table 1 not available, please ensure it

In line 195 , 12 in bracket probably mistakenly written. Please have an opinion on it

Ref :

Ref 42: Please mention journal name

Ref 60 : Page number missing

Reviewer #2: This manuscript describes a secondary analysis of an observational study of NIRS data collected within the first 15 minutes of life from term and preterm infants delivered by cesarean section. The investigators have correlated NIRS data with blood gas data at a similar time point. These investigators have published extensively on this topic and have contributed significant knowledge about cerebral oxygenation and transitional physiology previously. The findings presented in this manuscript are however a bit unclear.

1. The language used in the manuscript suggests that specific values on the blood gas or lactate levels cause changes in the cerebral tissue oxygenation or FTOE. Since the study is evaluating correlation alone there is no way to determine whether there is a causative relationship between these factors. Even if a causative relationship were known, the correlation cannot determine which factor influences the other. For example a higher lactate is associated with lower regional oxygen values. From that information alone one could not know whether the higher lactate caused the lower oxygen levels or whether the lower oxygen levels caused the higher lactate level.

2. The investigators hypothesized that there would be a correlation between blood gas and lactate levels and cerebral oxygen measurements using NIRS. They do not hypothesize what type of relationship they expected or how that relationship would be interpreted. Ultimately they found a relationship that seems unexpected in term infants with HCO3 positively correlated with FTOE. They have attempted to explain this relationship but haven't really made a complete thoughtful argument about what the relationship means physiologically. A really thoughtful discussion about the interpretation of these data would be critical to make this a useful contribution to the current state of knowledge.

3. The associations described in the manuscript that are statistically significant are of fairly low levels of correlation. The authors should consider that some of these finding may have occurred by chance especially since many comparisons were performed.

4. The authors state that the data were obtained from a previously performed 2-center observational study. The reference cited about that study was from a study of transcutaneous CO2 monitoring that doesn't mention NIRS use and only took place at one of the centers. Was this the correct reference? I didn't find another of the authors prior publications that met these same inclusion criteria.

5. The methods describe use of the NIRS monitoring but they don't state how long monitoring was kept in place or whether the resuscitation team was instructed to intervene based on the NIRS values.

6. PLOS authors have the option to publish the peer review history of their article (what does this mean?). If published, this will include your full peer review and any attached files.

Reviewer #1: No

Reviewer #2: No

---

## [Author Response · Author response to Decision Letter 0]

23 Sep 2022

Editors comments:

Response: Was reviewed

Response: The form has been corrected

GP is a recipient of the Culture Department of the City of Graz, Austria. 

GMS is a recipient of the Heart and Stroke Foundation/University of Alberta Professorship of Neonatal Resuscitation, a National New Investigator of the Heart and Stroke Foundation Canada and an Alberta New Investigator of the Heart and Stroke Foundation Alberta. This research has been facilitated by the Women and Children’s Health Research Institute through the generous support of the Stollery Children's Hospital Foundation.

Response: The form has been corrected

Response: Has been implemented

Response to the Reviewer:

Reviewer #1:

It seems to be a new area to explore impact of acid-base and other metabolic parameters on cerebral oxygenation in preterm and term neonates immediately after birth. Hence I have some observation.

Method:

- Required sample size better to mention.

Response: As already mentioned in the methods this was a post-hoc analyses of secondary outcome parameters of two prospective observational studies, therefore, no sample size calculation was not applicable.

Result:

- Table 1 not available, please ensure it

Response: Table 1 is uploaded and should be available (Results, Page 10, Line 186) 

- In line 195 , 12 in bracket probably mistakenly written. Please have an opinion on it

Response: (12) is the citation for published reference ranges for cerebral oxygenation - this is cited more clearly. (Results, Page 10, Line 197)

Ref 42: Please mention journal name

Response: Thank you. Was corrected. (References, Page 23, Line 514)

Ref 60 : Page number missing

Response: Thank you. Was corrected. (References, Page 24, Line 559-560)

 

Reviewer #2: 

This manuscript describes a secondary analysis of an observational study of NIRS data collected within the first 15 minutes of life from term and preterm infants delivered by cesarean section. The investigators have correlated NIRS data with blood gas data at a similar time point. These investigators have published extensively on this topic and have contributed significant knowledge about cerebral oxygenation and transitional physiology previously. The findings presented in this manuscript are however a bit unclear.

1. The language used in the manuscript suggests that specific values on the blood gas or lactate levels cause changes in the cerebral tissue oxygenation or FTOE. Since the study is evaluating correlation alone there is no way to determine whether there is a causative relationship between these factors. Even if a causative relationship were known, the correlation cannot determine which factor influences the other. For example a higher lactate is associated with lower regional oxygen values. From that information alone one could not know whether the higher lactate caused the lower oxygen levels or whether the lower oxygen levels caused the higher lactate level.

Response: Reviewer #2 is right that the present study did not prove causative relations but demonstrated “associations”. Therefore the authors concluded in the abstract that “There were significant associations… “ and concluded in the main text that “…acid-base and metabolic parameters are significantly associated …” and that “…the causal relationship between acid-base and metabolic parameters and cerebral oxygenation… have to be evaluated.”

2. The investigators hypothesized that there would be a correlation between blood gas and lactate levels and cerebral oxygen measurements using NIRS. They do not hypothesize what type of relationship they expected or how that relationship would be interpreted. 

Response: Thanks the reviewer for this comment. The hypothesis is now mentioned more detailed and clearly: “We hypothesized that higher LAC and lower pH, BE and HCO3 suggesting impaired metabolism is associated with lower crSO2 and higher FTOE values in preterm and term neonates 15 minutes after birth.” (Introduction, Page 5, Line 104-107)

Ultimately they found a relationship that seems unexpected in term infants with HCO3 positively correlated with FTOE. They have attempted to explain this relationship but haven't really made a complete thoughtful argument about what the relationship means physiologically. A really thoughtful discussion about the interpretation of these data would be critical to make this a useful contribution to the current state of knowledge.

Response: The current state of knowledge in relation to a correlation between HCO3 and the cerebral oxygenation in neonates is controversial. We observed a positive correlation between FTOE and HCO3 in term neonates 15 minutes after birth. Furthermore, we observed a trend towards a negative correlation between FTOE and HCO3 in preterm neonates. As an explanation the authors suggest – as already mentioned in the discussion- the strong link between the pH and HCO3.

The present study supports the strong link between pH and HCO3 which is mentioned now more clearly. (Discussion Page 18 line 361) Further suggestion or explanations are lacking but the present finding demonstrating the different associations for the first time suggest necessity of further physiological studies to give clear explanations as mentioned in the conclusion. 

3. The associations described in the manuscript that are statistically significant are of fairly low levels of correlation. The authors should consider that some of these finding may have occurred by chance especially since many comparisons were performed.

Response: Reviewer #2 is right. We mention this already in the limitation section (Discussion, Page 18, Line 375-377). In addition we mention now the methods: “The correlation analyses were considered in an explorative sense; therefore, no multiple testing corrections were performed.” (Methods, Page 8, Line 160)

4. The authors state that the data were obtained from a previously performed 2-center observational study. The reference cited about that study was from a study of transcutaneous CO2 monitoring that doesn't mention NIRS use and only took place at one of the centers. Was this the correct reference? I didn't find another of the authors prior publications that met these same inclusion criteria.

Response: The present post-hoc analysis was not performed with data from one previously performed 2-center study but from 2 studies performed at 2 centres – this is mentioned now more clearly in the methods section (Material and Methods, Page 6, Line 111)

5. The methods describe use of the NIRS monitoring but they don't state how long monitoring was kept in place or whether the resuscitation team was instructed to intervene based on the NIRS values.

Response: The NIRS monitoring was continuously during the first 15 minutes after birth. This is mentioned now more clearly (Material and Methods, Page 7, Line 138). For the present study we analyzed the 15th minute after birth. This has already been mentioned. (Material and Methods, Page 7, Line 142).

The present studies were observational studies as mentioned in the methods and not interventional studies. Thus, no interventions were based on NIRS monitoring. Resuscitation was performed by dedicated resuscitation teams, which were not involved in the study as already mentioned in the methods (Material and Methods, Page 6, Line 128-130).

---

## [Decision Letter · Decision Letter 1]

22 Dec 2022

PONE-D-22-00810R1Acid-base and metabolic parameters and cerebral oxygenation during the immediate transition after birth - A two-center observational studyPLOS ONE

Dear Dr. Pichler,

Thank you for submitting your manuscript to PLOS ONE. After careful consideration, we feel that it has merit and has important improvements compared to the original submission but does not fully meet PLOS ONE’s publication criteria as it currently stands. Therefore, we invite you to submit a revised version of the manuscript that addresses the points raised during the review process.

To meet the requirements for review I invited a third reviewer as reviewer 2 did not respond to my re-invitation. Reviewer 3 raises specific points that are suited to further clarify the presented results. Therefore, I kindly ask you to carefully address the points raised.

We look forward to receiving your revised manuscript.

Kind regards,

Harald Ehrhardt

Academic Editor

PLOS ONE

Reviewers' comments:

Reviewer's Responses to Questions

**Comments to the Author**

1. If the authors have adequately addressed your comments raised in a previous round of review and you feel that this manuscript is now acceptable for publication, you may indicate that here to bypass the “Comments to the Author” section, enter your conflict of interest statement in the “Confidential to Editor” section, and submit your "Accept" recommendation.

Reviewer #1: (No Response)

Reviewer #3: (No Response)

2. Is the manuscript technically sound, and do the data support the conclusions?

Reviewer #1: (No Response)

Reviewer #3: Yes

3. Has the statistical analysis been performed appropriately and rigorously? 

Reviewer #1: (No Response)

Reviewer #3: Yes

4. Have the authors made all data underlying the findings in their manuscript fully available?

Reviewer #1: (No Response)

Reviewer #3: Yes

5. Is the manuscript presented in an intelligible fashion and written in standard English?

Reviewer #1: (No Response)

Reviewer #3: Yes

6. Review Comments to the Author

Reviewer #1: (No Response)

Reviewer #3: Thank you for the opportunity to review this manuscript revision entitled, " Acid-base and metabolic parameters and cerebral oxygenation during the immediate transition after birth - A two center observational study". This study is a secondary analysis of a previously published observational study. The authors explore the association between cerebral saturation/FTOE and metabolic/pH parameters. My comments are below:

1. The introduction – especially the first paragraph - could be made much more concise.

2. What was the rationale for only using the average of the 15th min in the analysis – why not a longer window, especially given this time of transition. Also please explicitly state if the NIRS values were blinded during the study.

3. Given the low levels of correlation, I feel that this would have been strengthened by multiple regression analysis to account for size, respiratory support, etc. It also would be important to know the SaO2, FiO2, and CO2 at the time of the gas as well. When discussing cerebral blood flow and pH one would be remiss not to include CO2.

4. The discussion (parts with sub-headings) reads as a restatement of the results with listing various articles – the explanation interpretation feels lacking in these paragraphs – the paragraphs starting on page 16 line 309 should be incorporated into each of the above sections to allow for readability and data interpretation.

5. Were cord gasses available? I ask as long-standing vs more acute metabolic derangements may play a factor in the regulatory mechanisms.

6. I feel the clinical importance of this is not clear – what do these associations mean to the clinician, what would you target, etc.

7. PLOS authors have the option to publish the peer review history of their article (what does this mean?). If published, this will include your full peer review and any attached files.

Reviewer #1: No

Reviewer #3: No

---

## [Author Response · Author response to Decision Letter 1]

10 Feb 2023

Thank you very much for reviewing our manuscript “Acid-base and metabolic parameters and cerebral oxygenation during the immediate transition after birth - A two-center observational study” for publication in PLOS ONE. We have made the following changes according to the three Reviewer’s suggestions:

POINT TO POINT RESPONSE

Editors comments:

Response: Was reviewed

Response: The form has been corrected

GP is a recipient of the Culture Department of the City of Graz, Austria. 

GMS is a recipient of the Heart and Stroke Foundation/University of Alberta Professorship of Neonatal Resuscitation, a National New Investigator of the Heart and Stroke Foundation Canada and an Alberta New Investigator of the Heart and Stroke Foundation Alberta. This research has been facilitated by the Women and Children’s Health Research Institute through the generous support of the Stollery Children's Hospital Foundation.

Response: The form has been corrected

Response: Has been implemented

Response to the Reviewer:

Reviewer #1:

It seems to be a new area to explore impact of acid-base and other metabolic parameters on cerebral oxygenation in preterm and term neonates immediately after birth. Hence I have some observation.

Method:

- Required sample size better to mention.

Response: As already mentioned in the methods this was a post-hoc analyses of secondary outcome parameters of two prospective observational studies, therefore, sample size calculation was not applicable.

Result:

- Table 1 not available, please ensure it

Response: Table 1 is uploaded and should be available (Results, Page 9, Line 184) 

- In line 195, 12 in bracket probably mistakenly written. Please have an opinion on it

Response: (12) is the citation for published reference ranges for cerebral oxygenation - this is cited now more clearly. (Results, Page 9, Line 195)

Ref 42: Please mention journal name

Response: Thank you. Was corrected. (References, Page 21, Line 501)

Ref 60 : Page number missing

Response: Thank you. Was corrected. (References, Page 22, Line 545-547)

 

Reviewer #2: 

This manuscript describes a secondary analysis of an observational study of NIRS data collected within the first 15 minutes of life from term and preterm infants delivered by cesarean section. The investigators have correlated NIRS data with blood gas data at a similar time point. These investigators have published extensively on this topic and have contributed significant knowledge about cerebral oxygenation and transitional physiology previously. The findings presented in this manuscript are however a bit unclear.

1. The language used in the manuscript suggests that specific values on the blood gas or lactate levels cause changes in the cerebral tissue oxygenation or FTOE. Since the study is evaluating correlation alone there is no way to determine whether there is a causative relationship between these factors. Even if a causative relationship were known, the correlation cannot determine which factor influences the other. For example a higher lactate is associated with lower regional oxygen values. From that information alone one could not know whether the higher lactate caused the lower oxygen levels or whether the lower oxygen levels caused the higher lactate level.

Response: Reviewer #2 is right that the present study did not prove causative relations but demonstrated “associations”. Therefore the authors concluded in the abstract that “There were significant associations… “ and concluded in the main text that “…acid-base and metabolic parameters are significantly associated …” and that “…the causal relationship between acid-base and metabolic parameters and cerebral oxygenation… have to be evaluated.”

2. The investigators hypothesized that there would be a correlation between blood gas and lactate levels and cerebral oxygen measurements using NIRS. They do not hypothesize what type of relationship they expected or how that relationship would be interpreted. 

Response: Thanks the reviewer for this comment. The hypothesis is now mentioned more detailed and clearly: “We hypothesized that higher LAC and lower pH, BE and HCO3 suggesting impaired metabolism is associated with lower crSO2 and higher FTOE values in preterm and term neonates 15 minutes after birth.” (Introduction, Page 5, Line 105-107)

Ultimately they found a relationship that seems unexpected in term infants with HCO3 positively correlated with FTOE. They have attempted to explain this relationship but haven't really made a complete thoughtful argument about what the relationship means physiologically. A really thoughtful discussion about the interpretation of these data would be critical to make this a useful contribution to the current state of knowledge.

Response: The current state of knowledge in relation to a correlation between HCO3 and the cerebral oxygenation in neonates is controversial. We observed a positive correlation between FTOE and HCO3 in term neonates 15 minutes after birth. Furthermore, we observed a trend towards a negative correlation between FTOE and HCO3 in preterm neonates. As an explanation the authors suggest – as already mentioned in the discussion- the strong link between the pH and HCO3.

The present study supports the strong link between pH and HCO3 which is mentioned now more clearly. (Discussion Page 16 line 347-348) Further suggestion or explanations are lacking but the present finding demonstrating the different associations for the first time suggest necessity of further physiological studies to give clear explanations as mentioned in the conclusion. 

3. The associations described in the manuscript that are statistically significant are of fairly low levels of correlation. The authors should consider that some of these finding may have occurred by chance especially since many comparisons were performed.

Response: Reviewer #2 is right. We mention this already in the limitation section (Discussion, Page 17, Line 363-365). In addition we mention now the methods: “The correlation analyses were considered in an explorative sense; therefore, no multiple testing corrections were performed.” (Methods, Page 7, Line 157-158)

4. The authors state that the data were obtained from a previously performed 2-center observational study. The reference cited about that study was from a study of transcutaneous CO2 monitoring that doesn't mention NIRS use and only took place at one of the centers. Was this the correct reference? I didn't find another of the authors prior publications that met these same inclusion criteria.

Response: The present post-hoc analysis was not performed with data from one previously performed 2-center study but from 2 studies performed at 2 centres – this is mentioned now more clearly in the methods section (Material and Methods, Page 5, Line 109-111)

5. The methods describe use of the NIRS monitoring but they don't state how long monitoring was kept in place or whether the resuscitation team was instructed to intervene based on the NIRS values.

Response: The NIRS monitoring was continuously during the first 15 minutes after birth. This is mentioned now more clearly (Material and Methods, Page 7, Line 137-138). For the present study we analyzed the 15th minute after birth. This has already been mentioned. (Material and Methods, Page 7, Line 141-142).

The present studies were observational studies as mentioned in the methods and not interventional studies. Thus, no interventions were based on NIRS monitoring. Resuscitation was performed by dedicated resuscitation teams, which were not involved in the study as already mentioned in the methods (Material and Methods, Page 6, Line 127-129).

Reviewer #3:

This study is a secondary analysis of a previously published observational study. The authors explore the association between cerebral saturation/FTOE and metabolic/pH parameters. My comments are below:

1. The introduction – especially the first paragraph - could be made much more concise.

Response: We think a further compaction of the introduction will lead in a reduction in quality and comprehensibility.

2. 

i. What was the rationale for only using the average of the 15th min in the analysis – why not a longer window, especially given this time of transition. 

Response: Aim of the study was to correlate results of the acid base status measured once to cerebral oxygenation measured also once. Therefore the 15 minute was chosen since it was closest to the mean blood sample times in the two groups minute 17 and minute 16. This now mentioned in the methods: “The 15th minute median values of each neonate were used for analyses to be closest to the mean blood sample times in the two groups.” (Material and Methods, Page 7, Line 141-142)

ii. Also please explicitly state if the NIRS values were blinded during the study.

Response: Our study analyzed parameters of two prospective observational studies. Following this, no intervention was defined or was based on NIRS measurements and therefore blinding of NIRS was not performed. 

3. 

i. Given the low levels of correlation, I feel that this would have been strengthened by multiple regression analysis to account for size, respiratory support, etc. 

Response: In the present observational study a post-hoc analyses was performed and correlations were performed in an explorative sense – this is now mentioned in the methods: “The correlation analyses were considered in an explorative sense; therefore, no multiple testing corrections were performed.” (Material and Methods, Page 7, Line 157-158). 

Multiple regressions is difficult since there are a many influencing factors on cerebral oxygenation (oxygen content, perfusion consumption) and as the present study highlights metabolism.

ii. It also would be important to know the SaO2, FiO2, and CO2 at the time of the gas as well. When discussing cerebral blood flow and pH one would be remiss not to include CO2.

Response: Data are presented in table 1.

4. The discussion (parts with sub-headings) reads as a restatement of the results with listing various articles – the explanation interpretation feels lacking in these paragraphs – the paragraphs starting on page 16 line 309 should be incorporated into each of the above sections to allow for readability and data interpretation.

Response: Reviewer #3 has right. Thanks for this comment. For an increase in readability, we made following chances:

i. “A possible explanation of the observed negative correlation between crSO2 and LAC is …” (Discussion, Page 12-13, Line 252-260)

ii. “A possible explanation of the observed positive correlation between crSO2 and pH in …” (Discussion, Page 14-15, Line 285-310)

iii. “A possible explanations of our observed positive correlation between BE and crSO2 in preterm neonates are …” (Discussion, Page 15-16, Line 326-331) 

iv. “The present observations are in accordance with the findings in pH and crSO2 and FTOE in term and preterm neonates, since pH and HCO3 are strongly linked together…” (Discussion, Page 16, Line 347-348)

5. Were cord gasses available? I ask as long-standing vs more acute metabolic derangements may play a factor in the regulatory mechanisms.

Response: Thanks to the reviewer for this comment. In the present analyses cord gasses were not analyzed but we plan to investigate also possible associations between cord blood gases but still discussing which time point of cerebral oxygenation might be related best to cord blood gases. 

6. I feel the clinical importance of this is not clear – what do these associations mean to the clinician, what would you target, etc.

Response: The currently recommended monitoring for the assessment of preterm and term neonates after birth includes invasive and non-invasive monitoring technics. Association between non-invasive monitoring like, parameters of the pulse oximetry [peripheral arterial oxygen saturation (SpO2)] and/or electrocardiogram [heart rate (HR)] and the cerebral oxygenation in neonates during the immediate transition after birth are of increasing interest. Aim of the present study was to investigate if invasive measured variables like parameters of blood gas analysis (acid-base and other metabolic parameters) – parameters of metabolism are correlated to the cerebral oxygenation of preterm and term neonates during the immediate transition after birth. This knowledge can increase our understanding of the cerebral autoregulation mechanism even during the immediate transition and may lead in future to increase our possibilities in monitoring and intervention of neonates immediately after birth.

---

## [Decision Letter · Decision Letter 2]

7 Mar 2023

Acid-base and metabolic parameters and cerebral oxygenation during the immediate transition after birth - A two-center observational study

PONE-D-22-00810R2

Dear Dr. Pichler,

We’re pleased to inform you that your manuscript has been judged scientifically suitable for publication and will be formally accepted for publication once it meets all outstanding technical requirements.

Kind regards,

Harald Ehrhardt

Academic Editor

PLOS ONE

Additional Editor Comments (optional):

Reviewers' comments:

Reviewer's Responses to Questions

**Comments to the Author**

1. If the authors have adequately addressed your comments raised in a previous round of review and you feel that this manuscript is now acceptable for publication, you may indicate that here to bypass the “Comments to the Author” section, enter your conflict of interest statement in the “Confidential to Editor” section, and submit your "Accept" recommendation.

Reviewer #1: All comments have been addressed

Reviewer #3: All comments have been addressed

2. Is the manuscript technically sound, and do the data support the conclusions?

Reviewer #1: Yes

Reviewer #3: Yes

3. Has the statistical analysis been performed appropriately and rigorously? 

Reviewer #1: Yes

Reviewer #3: Yes

4. Have the authors made all data underlying the findings in their manuscript fully available?

Reviewer #1: Yes

Reviewer #3: Yes

5. Is the manuscript presented in an intelligible fashion and written in standard English?

Reviewer #1: Yes

Reviewer #3: Yes

6. Review Comments to the Author

Reviewer #1: (No Response)

Reviewer #3: (No Response)

7. PLOS authors have the option to publish the peer review history of their article (what does this mean?). If published, this will include your full peer review and any attached files.

Reviewer #1: No

Reviewer #3: No

---

## [Editor Report · Acceptance letter]

13 Mar 2023

PONE-D-22-00810R2 

Acid-base and metabolic parameters and cerebral oxygenation during the immediate transition after birth - A two-center observational study 

Dear Dr. Pichler:

I'm pleased to inform you that your manuscript has been deemed suitable for publication in PLOS ONE. Congratulations! Your manuscript is now with our production department. 

Kind regards, 

on behalf of

Prof. Harald Ehrhardt 

Academic Editor

PLOS ONE